# Beyond the Traditional: A Systematic Review of Digital Game-Based Assessment for Students' Knowledge, Skills, and Affections

Sha Zhu [1], Qing Guo [2] and Harrison Hao Yang [1,3,*]

1 National Engineering Research Center for E-Learning, Central China Normal University, Wuhan 430079, China
2 Faculty of Artificial Intelligence in Education, Central China Normal University, Wuhan 430079, China
3 School of Education, State University of New York at Oswego, Oswego, NY 13126, USA
* Correspondence: harrison.yang@oswego.edu

**Abstract:** Traditional methods of student assessment (SA) include self-reported surveys, standardized tests, etc. These methods are widely regarded by researchers as inducing test anxiety. They also ignore students' thinking processes and are not applicable to the assessment of higher-order skills. Digital game-based assessment (DGBA) is thought to address the shortcomings of traditional assessment methods. Given the advantages of DGBA, an increasing number of empirical studies are working to apply digital games for SA. However, there is a lack of any systematic review of DGBA studies. In particular, very little is known about the characteristics of the games, the content of the assessment, the methods of implementation, and the distribution of the results. This study examined the characteristics of DGBA studies, and the adopted games on SA in the past decade from different perspectives. A rigorous systematic review process was adopted in this study. First, the Web of Science (WOS) database was used to search the literature on DGBA published over the last decade. Then, 50 studies on SA were selected for subsequent analysis according to the inclusion and exclusion criteria. The results of this study found that DGBA has attracted the attention of researchers around the world. The participants of the DGBA studies were distributed across different educational levels, but the number of participants was small. Among all game genres, educational games were the most frequently used. Disciplinary knowledge is the most popular SA research content. Formative assessment modeling with process data and summative assessment using final scores were the most popular assessment methods. Correlation analysis was the most popular analysis method to verify the effectiveness of games on SA. However, many DGBA studies have reported unsatisfactory data analysis results. For the above findings, this study further discussed the reasons, as well as the meanings. In conclusion, this review showed the current status and gaps of DGBA in the SA application; directional references for future research of researchers and game designers are also provided.

**Keywords:** assessment methodologies; digital games; 21st century skills; media in education

## 1. Introduction

As a core component of psychometric and educational assessment, student assessment (SA) has long been an important research domain of interest to educational researchers. The aim of SA is to comprehend the degree to which students have acquired the knowledge, skills, and affections (KSAs) that are indispensable for full participation in contemporary societies [1]. Effective SA can inform instructors of potential instructional adjustments and additional scaffolding to provide tailored learning support for students [2–4]. Many national and international agencies have also launched extensive SA programs, including the Programme for International Student Assessment (PISA), the National Assessment of Educational Progress (NAEP), the Trends in International Mathematics and Science Study (TIMSS), etc. Up to now, many traditional assessment methods, such as self-reported

surveys and standardized tests, have been widely used in the field of SA [5]. However, these traditional methods suffer from subjective biases; they ignore the thinking process and are not applicable to the assessment of higher-order skills. In recent years, scholars and academia have widely accepted that SA should be integrated into complex life and learning contexts to stimulate authentic, fine-grained behavioral performance [6]. Digital game-based assessment (DGBA) enhances students' engagement and elicits authentic behaviors by constructing complex game contexts. As such, DGBA is considered a potential and viable alternative to traditional assessment methods [7].

## 1.1. Traditional SA Methods

The most prevalent SA methods are currently external measures, mainly including self-reported surveys and standardized tests [8]. The self-report survey is an indirect measure in which students are asked to evaluate how well they think they have performed certain tasks [9]. The traditional form of assessment known as a standardized test is given and scored in a predictable or standardized manner. Such tests have been considered to be a feasible method of direct SA [10]. Generally, a standardized test requires students to respond to pre-set questions in a paper or computer-supported format. This traditional method has mostly been confirmed by psychometric theory to be highly reliable and valid, making SA easy to implement on a large scale.

Still, however, the methodological shortcomings of these traditional tests and the consequent deficiencies of the assessment content cannot be ignored. In terms of methodology, the first difficulty to avoid is test anxiety, specifically because anxiety may affect students' motivation to engage [11], reduce working memory performance [12], and reinforce problem-solving difficulties [13]. The second problem is that traditional test methods can only determine students' final scores, but they cannot capture the students' thinking process. Actually, even if a student gives the wrong answer, their thinking and problem-solving process may still be appreciated [14]. The third problem is that students who are at a disadvantage on traditional tests might be from less privileged backgrounds, which potentially amplify inequalities.

In terms of content, the traditional methods involve deficiencies regarding the assessment of higher-order skills. In today's rapidly evolving world, 21st century skills have become indispensable for students to prepare for the future. Higher-order thinking skills, which encompass problem-solving, critical thinking, and creativity, are a critical component of these skills [15]. Policymakers, educators, researchers, and the general public have all recognized the importance of higher-order skills in empowering students to navigate complex challenges and succeed in their personal and professional lives [16]. Students with these skills are able to find answers and solve problems in real and confusing situations. However, the items presented from surveys and tests of traditional assessments are fixed and simplified, usually in textual format. The answers and responses to this type of material can hardly reflect the complex life context in which students grow up. As a result, the use of traditional assessment methods makes it difficult to stimulate and capture students' complex thinking and behavior. So, researchers believe that the traditional methods are applicable to the assessment of lower-order knowledge, rather than for gauging the higher-order skills required for survival in the 21st century [17].

## 1.2. Digital Game-Based Assessment

As the Organization for Economic Cooperation and Development (OECD) pointed out, the key to evaluation is not what students know, but what they can do with what they know [1]. Thus, in recent years, the focus of the student assessment domain has gradually shifted from declarative knowledge to higher-order skills related to practical life [18]. Traditional assessment methods are becoming increasingly resistant to the new requirements of SA, specifically due to the aforementioned deficiencies. Recently, it is generally accepted that SA should fit seamlessly into the fabric of complex learning environments and thus

be imperceptible to learners. Stealth assessment is the term for this idea [6], and DGBA is considered to be one of the key methods for achieving stealth assessment.

To be precise, DGBA is a special application of digital games in educational research, one which obtains accurate inferences about the extent of learners' KSA development by introducing game elements into the assessment structure [19]. Scholars maintain that DGBA has the potential to remedy the shortcomings of traditional assessment methods. First, DGBA creates interactive environments for students, an aspect that can largely eliminate their test anxiety. Additionally, DGBA contains the unique incentive mechanisms inherent in digital games, thereby having the capacity to enhance students' engagement and motivation [20]. Second, compared with traditional assessment methods, DGBA not only can obtain the outcomes of students' responses to game tasks, but can also capture the fine-grained process data generated during students' interactions with the game in a non-destructive way [8]. Analysis of these fine-grained data can uncover more details about students' cognitive abilities [21] and can provide students and instructors with dashboards that not only show students' final scores, but also reproduces the problem-solving process to support individualized learning and interventions [22]. Third, designing a game that is accessible and understandable to most students may somewhat enhance the equality among students. Fourth, digital games have permeated the lives of almost all students, and a systematic review study pointed out that digital game-based learning can promote the development of students' 21st century higher-order skills [23]. In terms of assessment, DGBA can create real-world interactive gaming environments that ensure the life fidelity of SA [24]. Within such assessment environments, students are able to demonstrate their complete and authentic thinking and engage in the process of complex problem-solving [18]. Thus, DGBA offers new possibilities for measuring SA, especially with regard to students' higher-order skills in the real world.

### 1.3. Previous Reviews and the Present Study

Given the aforementioned advantages, DGBA has recently been widely adopted in various fields. For example, many subjects, including biology [22], mathematics [25], science [14], and reading [26], utilize DGBA to measure students' knowledge acquisition. Many studies have also demonstrated the effectiveness of DGBA in assessing 21st century higher-order skills, such as social skills [27], creativity [28], and problem-solving [29]. Therefore, systematic and comprehensive reviews of DGBA studies are necessary. The review discussed here will provide an overview of the previous findings, point out the gaps in the literature, and offer suggestions for further research.

However, when searching for studies related to digital games in education, this study found that the vast majority of systematic reviews or meta-analysis studies only focused on the impact of game-based learning on specific academic fields. These studies reviewed the quasi-experimental process and the final effects of using digital games as pedagogical tools or interventions. For example, Acquah et al. analyzed 26 empirical studies that used digital games to assist students' second language learning. The study came to the overall conclusion that digital games are beneficial for language acquisition [30]. Tokac et al. conducted a meta-analysis of 24 works of game-based learning of mathematics education and found that digital games generally contribute to students' mathematical achievements [31].

This study has thus far identified only one systematic review related to DGBA, which was authored by Gomez et al. and published in July 2022. That study also noted that the authors had not found any studies that reviewed the literature on DGBA [7]. Gomez et al. analyzed the application sites, main purposes, knowledge areas, data samples, data analysis methods and reporting limitations of DGBA. However, referring to just one study that tried to review DGBA is not enough; at the very least, the following gaps need to be bridged: first, the review by Gomez et al. did not restrict the test participants; the DGBA review also still lacks any specific discussion of SA. Second, there is a lack of summary of the digital games referred to in DGBA studies, including the game genres, game platforms,

and whether or not the games are commercial. Third, the DGBA studies are unclear about which content areas of SA were examined and what specific assessment methods were used in those studies. This study believes that revealing the assessment content and methods is of great importance, as doing so can show whether the current DGBA studies have fully exploited the methodology and content advantages, compared with traditional assessment methods. Fourth, what kind of data analysis results reported by the existing DGBA studies are worthy of being analyzed as representing the validity of DGBA?

The present study is built on the basis of our previous short review [5]. The objective of this review is to fill in the gaps of previous review related to DGBA, and to make suggestions for future DGBA studies. In particular, we focus on the studies in which students were the subject of assessment. We further present a comprehensive overview of the current status and shortcomings of DGBA studies in terms of the distribution of participants, the characteristics of the games, the application of the assessment methods, and the results of the data analysis. Based on the findings of our review, recommendations for future DGBA studies are further proposed. Specifically, the present study would respond to the following research questions (RQ):

RQ1: What was the overview of the participants in the DGBA studies, including the country regions, number, and educational levels of the participants?

RQ2: What were the characteristics of the games used in the DGBA studies, including the game genres, game platforms, and commercial access to the games?

RQ3: What were the main assessment contents, and what assessment methods were used in the DGBA studies?

RQ4: What data analysis techniques were adopted, and what data analysis results were reported in the DGBA studies?

## 2. Method

The preferred reporting items for systematic reviews and meta-analyses (PRISMA) were used as the standard methodology for this study's systematic review of the literature. [32]. First, the database was selected, and a set of search terms was used for fixed queries. Second, a set of inclusion and exclusion criteria was identified, based on the research question. Then, the literature that met the criteria were progressively filtered based on the titles, abstracts, and full texts. Finally, the literature for the research questions were coded, and a synthesis analysis was performed.

### 2.1. Database and Search Terms

Web of Science (WOS) was chosen as the search database, specifically because WOS is an authoritative database covering all scientific citations [33]. In WOS, we retrieved more than 1500 relevant studies and finally screened 50 studies that met the inclusion criteria. These studies are sufficient for a systematic review to reveal the current status and gaps of studies in the field of DGBA. To guarantee that pertinent research would be included, the search terms "digital games" and "evaluation" were broadly designed in this study. The following were the specific search terms:

**Search phrases related to digital games:** "computer games" OR "video games" OR "serious games" OR "digital learning games" OR "digital education games" OR "digital games" OR "online games" OR "Internet games" OR "game-based" OR "game-assisted" OR "game-enhanced" OR "gameplay" OR "epistemic games".

**Search phrases related to assessment:** "assessment" OR "evaluation" OR "evaluate" OR "evaluating" OR "measure" OR "measuring" OR "measurement".

The *AND* operator was used to link the two search sets.

### 2.2. Inclusion and Exclusion Criteria

After obtaining the search results in the database, the inclusion and exclusion criteria were designed, as shown in Table 1. The aim was to make the selected studies eligible for the questions proposed in this study.

**Table 1.** The inclusion and exclusion criteria of this study.

| Inclusion Criteria | Exclusion Criteria |
| --- | --- |
| The research must be in the field of education or educational psychology. | All non-educational fields, such as medical, workplace, etc. |
| The study must be written in English. | Written not in English. |
| The use of digital games must be for the purpose of SA. | Other research purposes, such as game-based learning, game design, etc. |
| The participants of the study must be students. | The participants are not students, such as adults in the workplace, patients in medical settings, teachers, etc. |
| The study must have conducted an empirical inquiry using a digital game. | Other forms of research, such as framework proposal, qualitative study, case study, content analysis, etc. |
| The data collected were derived from the player's click interactions with the game. | Data obtained outside the game, such as questionnaires, physiological (such as eye movement) or neurological (such as electroencephalogram) data. |

*2.3. Study Selection*

In our previous brief review, the time interval was set from 1 January 2011 to 31 December 2021 [5]. For more comprehensive inclusion in this study, the search results from 1 January 2022 to 31 December 2022 were obtained using the same search terms. After two rounds of searching, a total of 1553 unique studies were found. Then, the first screening was performed by scanning the titles and abstracts of all the search results. In all, 1457 studies were excluded for not meeting the inclusion criteria. These studies offended at least one of the criteria by not belonging to the field of education or educational psychology, not being written in English or having been published in a non-English publication, not featuring assessment, or not involving students at the preschool to graduate level. Next, an additional full-text review of the remaining 96 studies was conducted in order to reconfirm the applicability of the studies. In the final screening, 46 studies were excluded for not meeting the inclusion criteria. Exclusion criteria included not conducting an empirical inquiry to obtain relevant data, or all the data collected were external from the game. A total of 50 studies made up the final sample, and Figure 1 depicts the entire selection procedure.

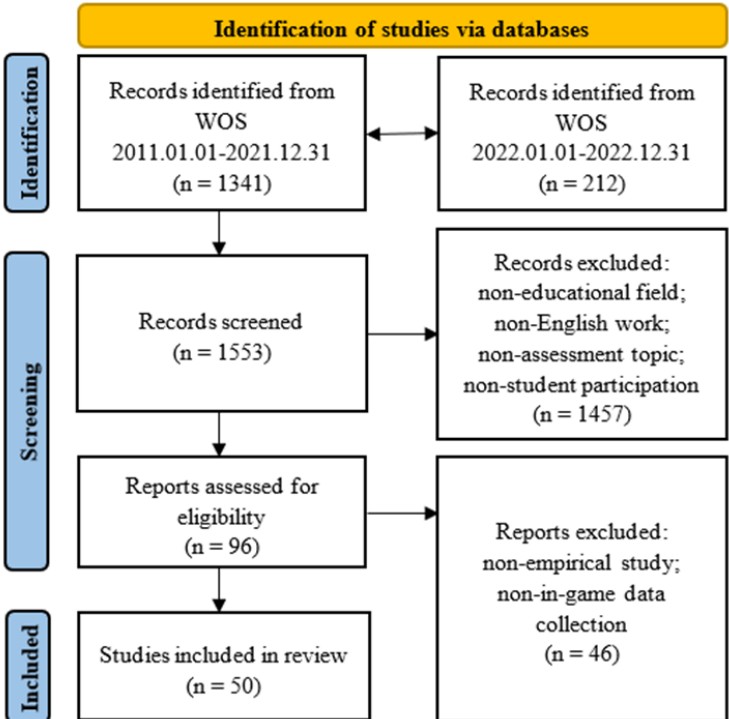

**Figure 1.** Study selection process for this systematic review.

*2.4. Data Analysis*

The first step of data analysis in this study was literature coding. An open coding scheme was followed, meaning that the codes were created in an open process, based on the content of the literature [7]. The coding process is iterative, and the coders can add or adjust the coding rules while they scrutinize the content of the literature. Specifically, this study first briefly reviewed all selected studies and developed preliminary coding rules for each RQ. Then, each piece of literature was carefully reviewed; the information corresponding to each RQ was checked, and the coding rules were iterated to finally complete the coding of all literature. It should be emphasized that each individual piece of literature can contain multiple codes created for each RQ. After completing the coding, statistical analysis was performed before reporting the results of each RQ.

## 3. Results

*3.1. Overview of the Participants in the Included Studies*

### 3.1.1. Participants' Country Regions

The continents of the countries where the study was conducted were counted. Some studies involved multiple participating countries, which were coded separately. There were 11 (20.8%) studies conducted in Asia, 12 (22.6%) in different parts of Europe, 24 (45.3%) in North America, three (5.7%) in South America, two (3.8%) in Africa, and one (1.9%) in Oceania. Figure 2 shows the individual countries and regions of the participants in the DGBA studies. The largest number of studies (22) were conducted in the United States (41.5%), followed by five (9.4%) studies in Spain, four (7.5%) studies in China, three (5.7%) studies in Finland, and one or two studies in each of the remaining countries.

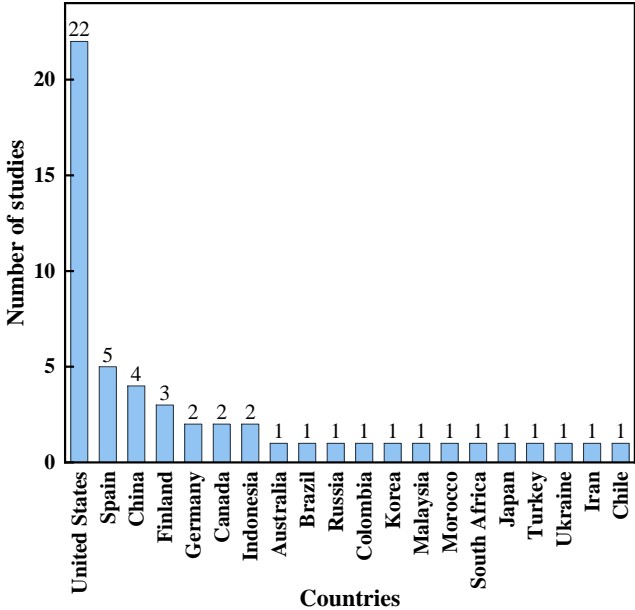

**Figure 2.** The number of studies in different countries.

### 3.1.2. Numbers of Participants

Figure 3 shows the distribution of the numbers of participants in DGBA studies. The bars in the figure represent the number of studies within a certain participants' range, and the curves represent the cumulative percentage of the number of studies. A few studies did not report a specific number of participants and were therefore ignored. Other studies had multiple sub-studies with multiple groups of participants and were, therefore, duplicate-coded. Finally, 23 (46.9%) studies with less than 100 participants were found, along with 15 (30.6%) studies with 100–200 participants. The cumulative percentage of studies with

less than 200 participants amounted to 77.5%. The median number of participants for all the studies was 102.

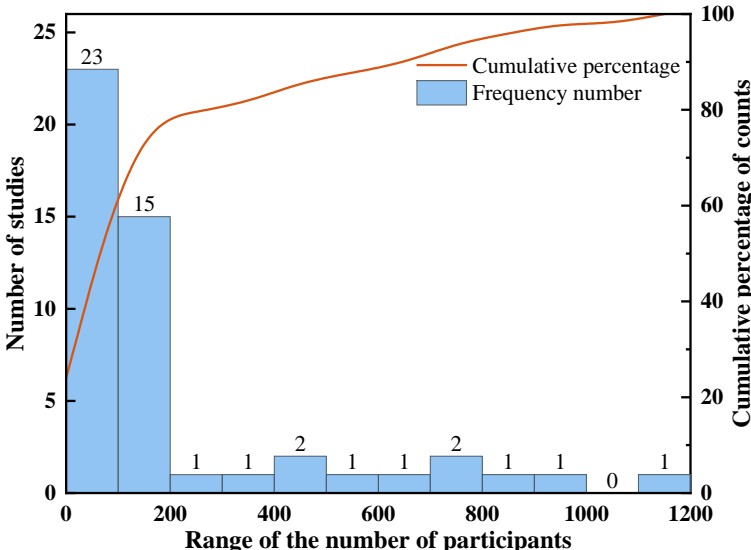

**Figure 3.** The number of studies within different participant range.

3.1.3. Educational Levels of the Participants

According to Figure 4, participants in the included DGBA studies were spread across a range of educational levels, with five (9.3%) studies involving preschoolers, 15 (27.8%) studies involving elementary schoolchildren, 22 (40.7%) studies involving middle school or high school students, and 12 (22.2%) studies involving college students. Overall, DGBA researchers focused more on K-12 students.

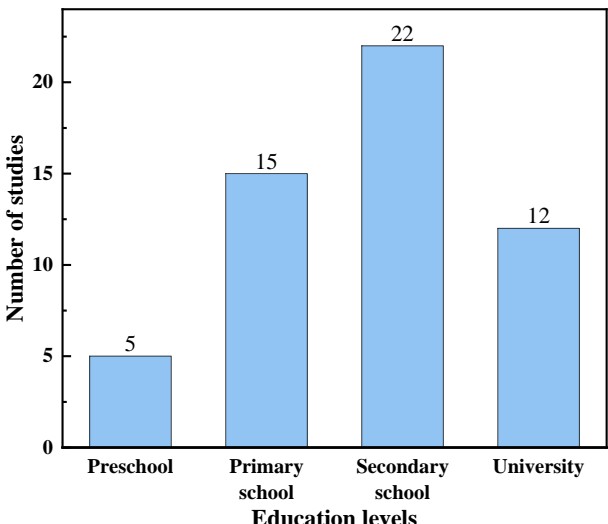

**Figure 4.** The number of studies at different education levels.

*3.2. Characteristics of the Games Used in the Included Studies*

3.2.1. Platforms of the Games

Of all the studies that reported game platforms, 22 (64.7%) games were played on computers, six (17.6%) games were played on i-Pads, and only one (2.9%) game was played on mobile phones. Additionally, three (8.8%) games required specialized equipment, and two (5.9%) games supported multiple devices.

### 3.2.2. Genres of the Games

Acquah et al. classified the digital games used for learning into six genres, and the present study refers to this classification [30]. The six genres are adventure, simulation, strategy, role-playing, educational, and puzzle. The definition of each genre is provided in Table 2, and two example studies for each game genre are also presented in the last column of the table.

**Table 2.** Game genres, definitions and example studies.

| Game Genre | Definition of Game Genre | Example Studies |
| --- | --- | --- |
| Adventure | Explore the unknown and resolve riddles using narrative hints. | Buford & O'Leary, 2015 [34]; Min et al., 2019 [4] |
| Simulation | Attempt to simulate as closely as possible a variety of real-world situations. | Slimani et al., 2018 [22]; Weiner & Sanchez, 2020 [24] |
| Strategy | Establish a setting that encourages complicated problem-solving and analysis while giving players complete freedom over how they interact with, manage, and employ game characters and things. | Krebs et al., 2020 [35]; Halverson & Owen, 2014 [36] |
| Role-playing | Provide players the opportunity to engage with the people in the game's scenario while taking on the roles of individuals living in a fictional world. | Irava et al., 2019 [37]; DeRosier & Thomas, 2018 [27] |
| Educational | The game was created for a specific discipline or subject, with clear traces of knowledge learning. | Kiili et al., 2018 [25]; Chen et al., 2020 [14] |
| Puzzle | Stimulate logic, sensitivity, etc. by mobilizing players' eyes, hands, and brains. | Delgado-Gómez et al., 2020 [38]; Chuang et al., 2015 [28] |

As indicated in Figure 5, educational games were the predominant game genre, appearing in 17 (34%) relevant studies. For example, in the game *Raging Skies*, players are required to complete a number of tasks based on their scientific meteorological expertise. The tasks include measuring wind speed and direction, describing airflow patterns and precipitation, etc [14]. Role-playing and puzzle games were the second most popular game genres, with each appearing in nine (18%) studies. *Hall of Heroes* is an example of role-playing games in which students enroll a superhero academy and interact with game characters to demonstrate their social skills [37]. *Running Raccoon*, a game with an "infinite run" motif where students have to continuously concentrate on guiding a raccoon over obstacles, is an illustration of a puzzle game [38]. In addition, six (12%) of the studies were about adventure games or strategy games. *Portal 2 Gf*, for instance, is a 26-room adventure game. In order to find and activate the exit in the game, students must look for puzzle pieces [34]. *Immune Defense*, a "tower defense" game, is an example of the strategy game. Students are given free rein to position, improve, and combine things in order to defend a "lifeform" from adversaries [35]. Only three (6%) of the DGBA studies' works featured simulation games, which had the lowest presence. One example of simulation games is *ELISA*, which provides a biological virtual lab where students can develop and evaluate their immunology skills [22].

### 3.2.3. Commercial Access to the Games

In addition to the abovementioned six game genres, games can also be divided into two categories, according to their public availability: self-developed by researchers and commercial off-the-shelf games. Self-developed games, which are created by researchers, are usually used only for research purposes and are not publicly available on any application or in stores. Commercial off-the-shelf games are usually available for download in some of the app stores. These games may not have been initially developed for educational purposes, but were relevant to certain aspects of students' KSAs and thus caught the attention of DGBA researchers.

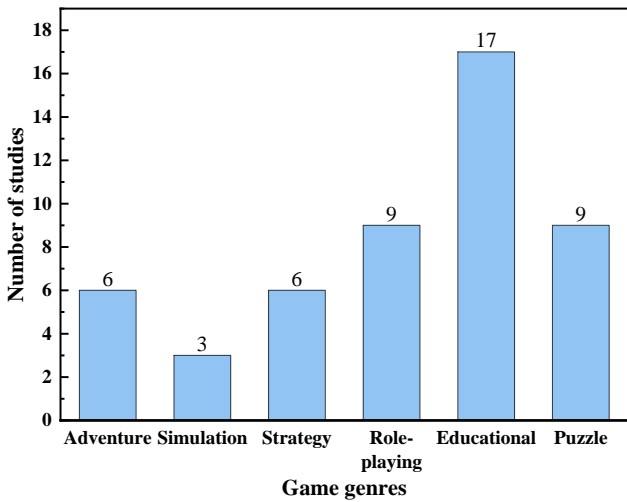

**Figure 5.** The number of studies of different game genres.

Five of the selected studies (10%) reported that they had used commercial off-the-shelf games. For example, Peters et al. measured students' intelligence levels using the game *Minecraft*, which is an open sandbox game [39]. The authors designed intellectual tasks and collected in-game data based on *Project Malmo*, an artificial intelligence experimental platform for *Minecraft*. *Project Malmo* gives users complete control over how the *Minecraft* environment is set up, including how the world is organized, what structures are used as targets, and how data is exported. Shute et al. accessed the source code of *Plants* vs. *Zombies* 2 and made minor modifications to create a new version with log data collection [29]. These data were used to assess students' problem-solving skills.

*3.3. Assessment Contents and Methods of the Included Studies*

3.3.1. Content of Assessment

The assessment content of DGBA studies has been classified into four categories here: discipline-specific knowledge, affective/psychological states, contemporary competencies and cognitive ability. The definitions and examples of each category are shown in Table 3.

**Table 3.** Definitions of the assessment content and example studies.

| Content of Assessment | Definition of the Assessment Content | Example Studies |
|---|---|---|
| Discipline-specific knowledge | Specific knowledge in a particular subject in school | Hautala et al., 2020 [26]; Kiili et al., 2018 [25] |
| Affective/psychological states | Involves attitude, awareness, perception, control, and emotion | Alonso-Fernández et al., 2020 [8]; Ventura & Shute, 2013 [40] |
| Contemporary competencies | High order skills necessary for students in the 21st century | Shute et al., 2016 [29]; Song & Sparks, 2019 [21] |
| Cognitive ability | The processing, storage and extraction of information by the human brain | Quiroga et al., 2015 [41]; Delgado-Gómez et al., 2020 [38] |

As shown in Figure 6, knowledge pertaining to specific disciplines was the main content of assessment, with 17 (34%) related studies involving biology, information science, mathematics, science, and reading. With six studies, accounting for 35.3% of the studies evaluating discipline-specific knowledge among these fields, mathematics was the most prominent. For example, Kiili et al. developed the *Semideus* game, which required students to complete calculations on a number line. The results were used to measure the students' knowledge of rational numbers in mathematics [25]. Cognitive abilities took second place, appearing in 15 (30%) of the studies. Specifically, the abilities being examined included intelligence, memory, attention, etc. For example, Delgado-Gómez et al. developed an infinite

running game named *Running Raccoon*, in which students are supposed to manipulate a raccoon to continuously cross obstacles. The object of the game is to demonstrate the students' attention levels [38]. Contemporary competencies were the next, appearing in 13 (26%) relevant studies. These competencies include argumentative skills, social skills, creativity, etc. For example, Song et al. developed the *Seaball* game to measure middle school students' argumentative skills. In the game, students had to debate certain social contextual issues [21]. Only five (10%) studies included assessments of affective/psychological states, being the lowest proportion. Additionally, the subjects covered a wide range, including persistence, autism, and other subjects. For example, Alonso-Fernández et al. assessed students' bullying awareness by using a role-playing game called *Conectado*. In that game, students enter a new school and are exposed to various forms of bullying [8].

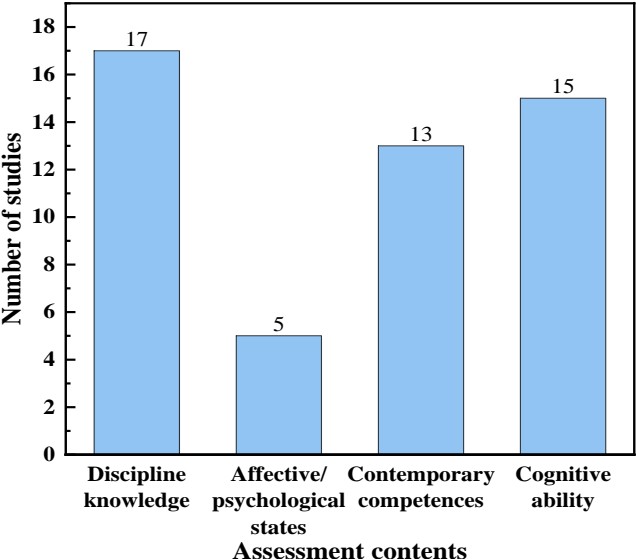

**Figure 6.** The number of studies with different assessment contents.

### 3.3.2. Method of assessment

This study groups the assessment methods into four categories based on their assessment orientation (summative vs. formative) and the type of data gathered (final game scores vs. process data created in game-playing): (1) summative assessment using final scores, (2) summative assessment using process data, (3) formative assessment using process data, and (4) formative assessment modeling with process data. The definitions and examples of each method are shown in Table 4.

**Table 4.** Definitions of the assessment method and example studies.

| Method of Assessment | Definition of the Assessment Method | Example Studies |
|---|---|---|
| Summative assessment using final scores | Use the game's final scores, including game coins and game score, as a gauge for the assessment content. | Song & Sparks, 2019 [21] <br> Wang et al., 2022 [42] |
| Summative assessment using process data | Utilize process information as indicators for direct assessment, such as playtime or the number of correct replies. | Kiili et al., 2018 [25]; <br> Tenorio Delgado et al., 2016 [43] |
| Formative assessment using process data | Calculate indicators based on the player's process data through the game's built-in formula. | Cutumisu et al., 2019 [44]; <br> Craig et al., 2015 [45] |
| Formative assessment modeling with process data | Mine feature variables using process data to build prediction models. | Chen et al., 2020 [14]; <br> Shute & Rahimi, 2021 [46] |

Some studies adopted multiple methods of assessment and were therefore duplicate-coded. As seen in Figure 7, the main assessment method was formative assessment modeling using process data, which was used in 19 (31.1%) of the sampled studies. These studies were mostly conducted using an evidence-centered design (ECD) framework, which comprises a student model for identifying the KSAs to be assessed, an evidence model for establishing connections between observable behavioral data and KSAs for modeling and forecasting, and a task model for devising tasks that elicit such behaviors [47]. An example is the study by Chen et al., which used machine learning models to analyze behavioral data from *Raging Skies* and predict students' acquisition of meteorological knowledge [14]. Another major method employed was summative assessment using final scores, which was applied in 18 (29.5%) studies. Twelve (19.7%) studies consisted of summative assessment using process data. Another 12 (19.7%) studies used process data for formative assessment. For instance, in the *ZOOU* game, a unique scoring algorithm was developed for each scene to accurately measure students' social skills by tracking and scoring their in-game behaviors [27].

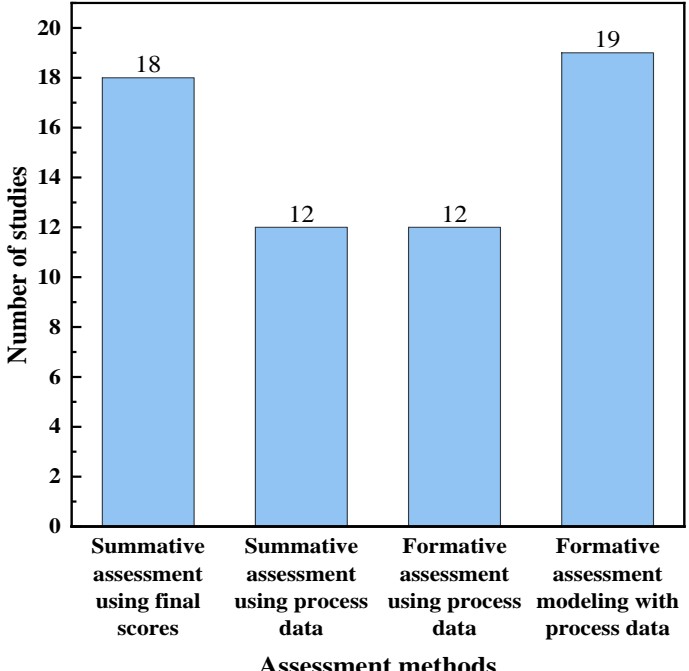

**Figure 7.** The number of studies and their different assessment methods.

*3.4. Data Analysis Techniques and Results of the Included Studies*

3.4.1. Data Analysis Techniques

The reviewed studies employed two principal categories of data analysis techniques, namely, supervised models and statistical analysis. Supervised models are included in machine learning modeling, in which students' KSAs are used as labels. In addition, the relationship between the in-game process data and the labels is 'trained' to achieve efficient predictions on new datasets. Statistical analysis refers to those techniques that are not modeled with KSAs labels, such as correlation analysis, difference tests, etc. In the studies included in this research, data analysis techniques were usually related to the assessment methods (see Table 5). Among the reviewed studies, all of those using supervised modeling techniques are in the fourth category of the abovementioned assessment methods, namely, formative assessment modeling with process data.

**Table 5.** Relationship between data analysis techniques and assessment methods.

| Method of Assessment | Number of Studies Using the Supervised Model Technique | Number of Studies Using the Statistical Analysis Technique |
|---|---|---|
| Summative assessment using final scores | 0 | 18 |
| Summative assessment using process data | 0 | 12 |
| Formative assessment using process data | 0 | 12 |
| Formative assessment modeling with process data | 18 | 11 |

Since many studies employed multiple data analysis techniques, they were duplicate-coded. The number of studies for each data analysis technique, as well as their percentage of the total number of studies, are shown in Table 6. A wide range of supervised models have been used, including regression and classification models, depending on the data style (continuous or categorical) of the KSAs labels. Linear regression was the most commonly used technique, having been adopted by five (10%) studies. This was followed by random forest and support vector machine, with each being adopted by four (8%) studies.

**Table 6.** Number of studies using each data analysis technique and their percentage of total studies.

| Data Analysis Technique | Number of Studies Using the Technique | Percentage of the Total Studies |
|---|---|---|
| **Supervised models** | | |
| Linear regression | 5 | 10% |
| Elastic net regression | 1 | 2% |
| Bayesian ridge regression | 1 | 2% |
| Mixed linear model | 1 | 2% |
| Logistic regression | 2 | 4% |
| K-nearest neighbor | 1 | 2% |
| Decision tree | 3 | 6% |
| Random forest | 4 | 8% |
| Gradient boosting decision tree | 2 | 4% |
| Adaboost | 1 | 2% |
| Support vector machine | 4 | 8% |
| Conditional random field | 1 | 2% |
| naïve bayes | 3 | 6% |
| Bayesian network | 2 | 4% |
| Dynamic Bayesian network | 1 | 2% |
| Deep neural network | 3 | 6% |
| Long short-term memory | 2 | 4% |
| **Statistical analysis** | | |
| Correlation analysis | 29 | 58% |
| Variance analysis | 11 | 22% |
| EM clustering | 2 | 4% |
| K-means clustering | 1 | 2% |

Correlation analysis was the most commonly used statistical analysis technique, being utilized in 29 studies, or a staggering 58% of the total number of studies. There are two different applications of correlation analysis. First, for data collected from the first three of the above assessment methods (final scores, process data, and indicators calculated from process data), the correlations between them and the external test results were calculated. A total of 21 studies took this correlation analysis approach, accounting for 72.4% of the total correlation analysis applications. Second, for the fourth category of assessment methods, eight (27.6%) studies that had built supervised models correlated the model predictions with the results of external tests.

### 3.4.2. Data Analysis Results

This study mainly focused on two data analysis results: the results reported by the supervised models and the coefficients of the correlation analysis, as these results

demonstrate the effectiveness of DGBA. When multiple analysis results were reported for the same indicator in a study, the best result was selected for coding. The analysis results of the studies that constructed the supervised models are shown in Table 7. For regression models, $R^2$ was below 0.5 overall, and the error indicators (MAE, RMSE) were above 0.5 overall. For the classification model, the reported accuracy was at minimum above 0.6 and at maximum up to 0.98. Other indicators, such as recall, sensitivity, specificity, and false positive rate were less reported. For the correlation coefficients between model predictions and external tests, four studies were below 0.5, three studies were below 0.7, and only one study reported a coefficient above 0.9.

**Table 7.** Results of data analysis for supervised modeling studies.

| Assessment Metric | Range of the Metric | Number of Studies |
| --- | --- | --- |
| $R^2$ | 0.260–0.431 | 4 |
| MAE | 0.540–0.640 | 2 |
| | >1 | 1 |
| RMSE | 0.506–0.770 | 2 |
| | >1 | 1 |
| Accuracy | 0.637–0.715 | 2 |
| | 0.900–0.980 | 2 |
| Recall | 0.980 | 1 |
| Sensitivity | 0.620 | 1 |
| | 0.925 | 1 |
| Specificity | 0.540 | 1 |
| False positive rate | 0.310 | 1 |
| | 0.203–0.410 | 4 |
| Correlation with external tests | 0.530–0.670 | 3 |
| | 0.970 | 1 |

For the correlation analysis results between the final game scores, process data, indicators calculated from process data, and external tests, a histogram was plotted with a cumulative percentage curve of the study counts (see Figure 8). As seen in the figure, the correlation coefficients show an overall normal distribution, with the largest number of studies between 0.5 and 0.6. There were 13 studies with coefficients lower than 0.6, accounting for 61.9%; only three studies had coefficients higher than 0.8, accounting for 14.3%.

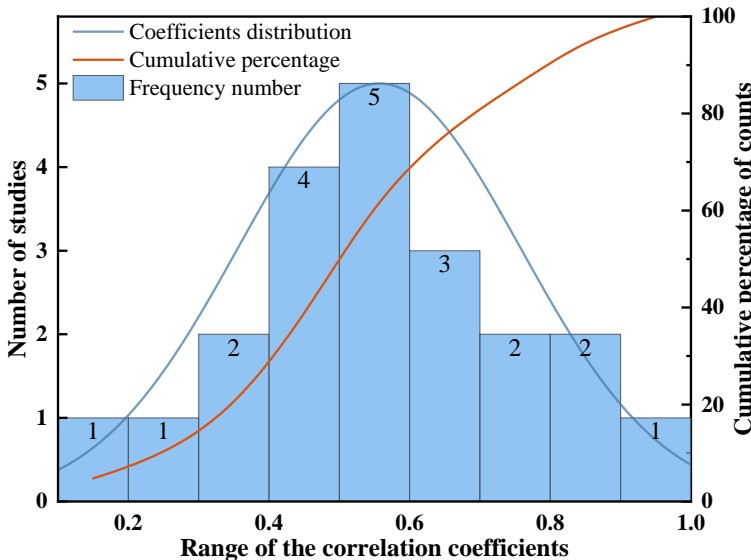

**Figure 8.** The histogram of the distribution of correlation analysis results.

## 4. Discussion

Today's students are expected to comprehensively develop KSAs, especially high-order skills in the 21st century. These skills are crucial for childrens' future success in school, the workplace, and in life [48]. Accurately assessing students' KSAs is a primary component of further cultivation. Traditional assessment methods suffer from some deficiencies, including the test anxiety problem, an ignorance of students' thinking processes, and an ineptitude for assessing higher-order skills. Today, DGBA is thought to be able to bridge these gaps. The strong interactivity and the motivation mechanism of DGBA diminish students' test anxiety; DGBA also supports the collection of fine-grained in-game interaction data, which can facilitate the understanding of cognitive and thinking processes. Furthermore, DGBA creates assessment environments in which students can play games in real-life situations. In turn, these real-life experiences appreciate and enhance the authentic performance of students in solving complex problems; thus, DGBA is suitable for the assessment of higher-order skills. Recently, there has been extensive attention and usage of DGBA in the SA domain, but there has also been a serious lack of any systematic review. This study reviews, in great detail, DGBA studies in the SA domain, including from the aspects of countries, participants, game characteristics, assessment contents, assessment methods, data analysis techniques, and assessment results. The results of the review are discussed below, and the recommendations for future research are also provided.

### 4.1. Current States of DGBA Studies

#### 4.1.1. The Overview of the Participants in the DGBA Studies

For RQ1, this study first found that DGBA research in the field of SA has covered six continents; this clearly shows growing interest from DGBA researchers at a global level. Of all continents, North America has conducted the most studies, followed by Europe and Asia. This study went further and counted the specific countries, and the studies conducted in the United States were multiple times higher than in other countries, accounting for more than 40% of the total number of studies. Although DGBA studies are widely distributed, the United States accounts for the majority, which proves that current DGBA research is inadequate and should attract the attention of researchers in more countries in the future.

The results of this study show that DGBA studies generally had fewer participants. Specifically, nearly 80% of the studies enrolled fewer than 200 participants, and only one study had more than 1000 participants. The median number of participants in all studies was only 102. This result is similar to a previous review of the application of data science methods to games. The authors of that review found that 32% of the studies engaged less than 100 participants, and only 8% of the studies exceeded 1000 participants [8]. This may be due to the fact that most of the current DGBA studies are exploratory, and researchers are more focused on validating the effectiveness of DGBA in small samples. This study also found that previous DGBA research covered almost all educational levels, with more in primary, middle and secondary schools. This finding is consistent with the educational levels upon which many large-scale student assessment programs focus, as the K-12 level is a critical stage of development for students' KSAs. We do not see this as a deficiency. It is important to focus on K-12 students, however we believe that the preschool level should be valued by future DGBA studies.

#### 4.1.2. The Characteristics of the Games Used in the DGBA Studies

For RQ2, according to the findings of this study, educational games have emerged as the most commonly used game genre in DGBA studies, which is consistent with Acquah's research [30]. This outcome is further corroborated by the observation that the assessment content has largely centered on discipline-specific knowledge. The structure of educational games is generally more well-defined, with a greater emphasis on educational content and a lower emphasis on entertainment value. Furthermore, when compared to other game genres, educational games are comparatively easier to develop. These characteristics make educational games a suitable choice for assessing students' discipline-related knowledge.

However, researchers have pointed out that a narrow focus on knowledge related to a specific discipline through simple game design may create difficulty in making the game appealing to students [23]. Among all game genres, the number of studies that adopted simulation games was the lowest. This may be due to the fact that simulation games require a high degree of reproduction of real-life scenarios, usually combined with virtual reality techniques. Such games are extremely difficult to develop.

One surprising finding is that most of the games were self-developed; very few studies used commercial off-the-shelf games. Commercial off-the-shelf games are more entertaining and engaging for students. The problem-solving process in the games is also more complex and can stimulate students' higher-order thinking. However, the problem with using commercial off-the-shelf games is the potential mismatch between the established game design and the student KSAs of interest to researchers [36].

### 4.1.3. The Assessment Contents and Methods Used in the DGBA Studies

For RQ3, echoing the largest proportion of educational games among the game genres discussed above, this study found that discipline-specific knowledge was the most popular assessment content for researchers. This may be attributed to the fact that assessment of knowledge of a particular discipline is the type most in demand in school education. In addition, educational games used to assess discipline-specific knowledge are less difficult to produce. Notably, an essential advantage of DGBA is the creation of authentic situations that stimulate complex problem-solving performance, with the game results used to measure students' higher-order skills. However, the shortcoming of previous DGBA studies is that a large portion of the studies focused on relatively low-order, discipline-specific knowledge. Therefore, the assessment of higher-order contemporary competencies needs to be reinforced to fully exploit the advantages of DGBA over traditional assessment methods.

This study reveals that the most popular approach to assessment in DGBA studies is formative assessment modeling with process data. This method is guided by ECD theory and involves constructing complex prediction models to achieve assessment purposes [49]. It allows for a detailed analysis of the gaming process, which fully utilizes the potential of stealth assessment to draw valid inferences about students' competencies and performance in a non-destructive manner [46]. Additionally, this method establishes a standardized, automated assessment procedure that simplifies the assessment of players in further game deployments [8]. On the other hand, summative assessment using final scores is another widely used method in DGBA studies. This approach typically and simply incorporates game elements into traditional tests and converts students' answers into game scores in lieu of test scores [50]. Although this method is easy to use and convenient to apply, the game content and scoring remain similar to traditional tests and do not change the orientation of summative assessment.

### 4.1.4. The Data Analysis Techniques and Results Reported in the DGBA Studies

For RQ4, in terms of those studies that used formative assessment modeling with process data, this review finds that the previous studies mostly constructed multiple supervised models, with linear regression, random forest, and support vector machine being the most popular algorithms. Another striking finding is that 58% of the studies adopted the technique of correlation analysis. The reason for this is that the general purpose of a DGBA study is to verify the validity of the game used to assess the content of interest. Therefore, the most common practice is to analyze the correlation between the final game scores, the process data, the indicators calculated from process data, or the predicted results of the supervised models and the results of the external tests' criterion.

The results of the data analysis were not highly satisfactory. In the supervised regression models, the $R^2$ indicator only reached a maximum of 0.431; the accuracy of the supervised classification models only reached a maximum of about 0.7, and half of the correlation coefficients between the prediction results of the supervised models and the external tests were below 0.5. The results of those studies (without supervised modeling)

showed that more than 60% of the correlation coefficients were below 0.6. One possible reason is that the results of traditional tests may be influenced by students' backgrounds. Digital games, however, are deeply integrated into most students' lives, and well-designed games might somewhat enhance the fairness of assessment, thus leading to differences in the results of the two methods. On the other hand, some DGBA studies have shown that transforming game data into constructs is a difficult procedure that typically yields low-performing results [51,52]. Regardless of the merits or demerits of the game design, most previous DGBA studies have been exploratory, and they aim to demonstrate the potential of DGBA to some extent. Therefore, researchers have considered that, although some studies had reported relatively low-performing results, most of these results were statistically significant. Additionally, these results are beneficial for future advances in the research field of DGBA [51].

### 4.2. Recommendations for Future Studies

This study first recommends that future DGBA studies should objectively use increased sample sizes, as small sample sizes may limit the significance and generalization of the results. Another suggestion is that future research should break the educational game bottleneck, take into account other game genres, and put more work into creating game situations that strike a balance between enjoyment and education. This is because for most students, entertainment is guaranteed to stimulate students' emotions and motivation, and is also a key to triggering the most authentic thinking processes and behavioral performance.

An essential recommendation for the assessment content is to strengthen the future study of higher-order skills in the 21st century. Undoubtedly, in this century, higher-order skills such as creativity, critical thinking, collaboration, problem solving, and communication have come to occupy the most important positions in student development. However, accurately assessing these new-age skills is a well-recognized challenge [23]. One of the major advantages of DGBA is the opportunity to create immersion and enhance interaction through well-designed game content. Additionally, a reward mechanism can be created to support the demonstration of students' performance of 21st century skills. Therefore, future research should place more emphasis on higher-order skills, rather than the relatively lower-order discipline-specific knowledge, so that the advantages of DGBA can be fully exploited.

With the future demand for the assessment of 21st century higher-order skills, this study strongly recommends that future studies adhere to the idea of evidence-based reasoning in assessment methods. This is precisely the approach taken by the studies that conducted formative assessment modeling with process data. Specifically, elaborate student models should be constructed to guide the development of assessment games. Reliable in-game evidence should be designed by high-proficiency subject experts, and automatic assessment models should be constructed to link the game evidences to the higher-order skills to be measured.

To improve the performance of the assessment results of DGBA, the following measures are recommended for future studies: first, as mentioned above, increase the sample size, elaborate games based on student models, and elaborate the evidence for reasoning. These are absolutely necessary initiatives. Second, data mining algorithms should also be widely used to obtain more stealthy feature variables from the game log data. Third, for behavioral sequences, time-series prediction models should be considered, as they can make full use of time-series information to potentially obtain excellent prediction results. Finally, game design for DGBA studies should be iterative, and satisfactory assessment results require multiple rounds of assessment implementation and tailored game improvements.

Although DGBA has been widely used, future studies should be alert to the potential disadvantages of applying digital games for SA. For example, students of different genders may have innate differences in gaming skills [53], and how to eliminate such gaps by improving game design or through instructional interventions is important for further consideration. Another issue is the gaming experience. Some students play games

frequently so that they can easily grasp the mechanisms and techniques of a new game and thus may outperform those with less experience [54]. Future studies should emphasize these individual differences related to game playing. Otherwise, DGBA, despite weakening students' backgrounds in SA, risks introducing new inequities.

*4.3. Limitation*

It is important to acknowledge the limitations of this study, including its exclusive focus on SA, while DGBA's potential applications in education may be more extensive. Future research should explore DGBA's applications in other educational assessment contexts. Additionally, this study did not explore the effects of continuous DGBA use. Longitudinal research is necessary to establish more robust evidence on the effectiveness of DGBA.

**5. Conclusions**

This study aims to investigate DGBA studies, and the games adopted for SA from various perspectives over the past decade. This comprehensive review presents surprising findings on the use of DGBA in the field of SA. The analysis reveals that DGBA has gained global attention from researchers, especially in the United States. The reviewed studies encompassed participants from various educational levels, with K-12 receiving the most attention. Nonetheless, the DGBA studies exhibited a tendency towards small sample sizes. Educational games, which are less entertaining, emerged as the most frequently used game genre. Disciplinary knowledge was the primary focus of assessment in the reviewed studies, as opposed to 21st century higher-order skills. The most common assessment methods included formative assessment modeling with process data and summative assessment using final scores. To validate the effectiveness of DGBA on SA, diverse data analysis methods were utilized, with correlational analysis being the most widespread. However, the results of data analysis from a considerable number of studies reported unsatisfactory efficacy of DGBA for SA. In conclusion, this study sheds light on the current status and gaps of DGBA studies in SA and offers directions for future research.

**Author Contributions:** Conceptualization, S.Z.; methodology, H.H.Y.; software, Q.G.; formal analysis, Q.G.; writing, Q.G.; writing—review and editing, S.Z. and H.H.Y.; supervision, S.Z. and H.H.Y. All authors have read and agreed to the published version of the manuscript.

**Funding:** The work was funded by a grant from the National Natural Science Foundation of China (No. 62107019).

**Institutional Review Board Statement:** Not applicable.

**Informed Consent Statement:** Not applicable.

**Data Availability Statement:** Not applicable.

**Conflicts of Interest:** The authors affirm that none of their known financial or personal affiliations might have seemed to affect the research presented in this paper.

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
