# Peer review of "Beyond the Traditional: A Systematic Review of Digital Game-Based Assessment for Students’ Knowledge, Skills, and Affections"

_sustainability, doi:10.3390/su15054693_

Round 1

Reviewer 1 Report

The assessment of higher-order thinking skills is critical for improving learning for students. This manuscript describes a rigorous examination of the literature. The four evaluation questions provide a rigorous examination of the merits of DGBA over reliance on SLA. The manuscript helps to fill the gap in the professional literature. The recommendations section identifies a road map for future research. I suggest moving the limitations section out of the conclusion and instead move it to just before it. I think that it deflects from the merits of your manuscript. End with a comprehensive, strong paragraph.

Considering what I observe from students in the U.S., I was surprised to see only one study that examined the impact of gaming on higher-order thinking skills. I was also surprised to see that commercial video games were not examined. Maybe it is as you said that few if any studies have considered commercial games and instead the others.

Your manuscript failed to convince me that gaming is a route to develop higher-order thinking skills. Also, I am not sure that I agree with your definition of 21st century higher-order thinking skills. Perhaps you could add some more paragraphs and references to convince me and your potential readers. It seemed there were too many assumptions made with these two arguments.

Author Response

Reviewer #1:      
The assessment of higher-order thinking skills is critical for improving learning for students. This manuscript describes a rigorous examination of the literature. The four evaluation questions provide a rigorous examination of the merits of DGBA over reliance on SLA.

  1. I suggest moving the limitations section out of the conclusion and instead move it to just before it. I think that it deflects from the merits of your manuscript. End with a comprehensive, strong paragraph.

Response: Thank you. According to your comments, the description of limitations in the conclusion section of the manuscript has been moved forward to the last point of the discussion section.

A new section 4.3 has been added to the discussion section, which was as follows:

4.3. Limitation

It is important to acknowledge the limitations of this study, including its exclusive focus on SA, while DGBA’s potential applications in education may be more extensive. Future research should explore DGBA’s applications in other educational assessment contexts. Additionally, this study did not explore the effects of continuous DGBA use. Longitudinal research is necessary to establish more robust evidence on the effectiveness of DGBA.

  1. Considering what I observe from students in the U.S., I was surprised to see only one study that examined the impact of gaming on higher-order thinking skills. I was also surprised to see that commercial video games were not examined. Maybe it is as you said that few if any studies have considered commercial games and instead the others.

Response: Thank you. In section 3.3.1 of the manuscript, we analyzed the assessment content that the DGBA studies focused on. As shown in Figure 6, the number of studies that used digital games to assess 21st century higher-order skills reached 13. With regard to commercial games, we believe that researchers may have difficulty obtaining access and modification of commercial games. The two cases of commercial game applications outlined in the manuscript both redeveloped the game content and data collection framework.

  1. Your manuscript failed to convince me that gaming is a route to develop higher-order thinking skills. Also, I am not sure that I agree with your definition of 21st century higher-order thinking skills. Perhaps you could add some more paragraphs and references to convince me and your potential readers. It seemed there were too many assumptions made with these two arguments.

Response: Thank you. Based on your comments, we further define higher-order skills in section 1.1 when describing the limitations of traditional assessment methods. We have added new content and cited references. Regarding the development of higher-order skills supported by games, we added descriptions and references when we elaborated on the advantages of DGBA in section 1.2.

The added definition in section 1.1 was as follows:

In today’s rapidly evolving world, 21st-century skills have become indispensable for students to prepare for the future. Higher-order thinking skills, which encompass problem-solving, critical thinking, and creativity, are a critical component of these skills [15]. Policymakers, educators, researchers, and the general public have all recognized the importance of higher-order skills in empowering students to navigate complex challenges and succeed in their personal and professional lives [16].  Students with these skills are able to find answers and solve problems in real and confusing situations. However, the items presented from surveys and tests of traditional assessments are fixed and simplified, usually in textual format. The answers and responses to this type of material can hardly reflect the complex life context in which students grow up. As a result, the use of traditional assessment methods makes it difficult to stimulate and capture students’ complex thinking and behavior.

The added descriptions and references in section 1.2 was as follows:

Fourth, digital games have permeated the lives of almost all students, and a systematic review study pointed out that digital game-based learning can promote the development of students’ 21st-century higher-order skills [23].

The new citations are as follows:

[15] Lu, K.; Yang, H.H.; Shi, Y.; Wang, X. Examining the key influencing factors on college students’ higher-order thinking skills in the smart classroom environment. International Journal of Educational Technology in Higher Education. 2021, 18, 1-13.

[16] Lu, K.; Yang, H.; Xue, H. Investigating the four-level inquiry continuum on college students’ higher order thinking and peer interaction tendencies. International Journal of Innovation and Learning. 2021, 30, 358-367.

[23] Qian, M; Clark, K.R. Game-based Learning and 21st century skills: A review of recent research. Computers in human behavior. 2016, 63, 50-58.

Reviewer 2 Report

Dear authors,

I read your article with interest. It is well organised, well written and insightful.

I have only a few minor comments:

1) the use of the word ‘literacy’ in the title and beginning of the text seems to me to be a bit misleading. Later in your sample you include studies that use games to assess KSAs that are only marginally related to what is meant by literacy. In other words, you seem to be describing your work as narrower in scope than it actually is.

2) I found the opening sentence of the introduction to be rather empty.

3) I believe 1.1. should be expanded. You make your argument vulnerable to the following objection: “sure, prevalent methods can’t capture a lot of interesting stuff, but they are simple, and the end result would not be very much different.” I think you need to preempt this objection for instance by noticing that students who are at a disadvantage in traditional tests are usually from less privileged backgrounds, which might amplify inequalities, or simply by bringing empirical literature to show that the results of using traditional methods are not highly correlated with the results of using the game based ones that you explore.

4) More should be said about the cons of using game based methods.

5) p. 4, l. 154 reads “First, the database and was selected”, so there is something missing.

6) you should clarify what a “non English publication” is.

7) p. 7, l. 259 reads “In reference to Acquah et al., this study finally classifies the games into six genres”. I found this sentence cryptic.

8) p. 15, ll. 513-518 reads “This is because entertainment is virtually guaranteed to stimulate students’ emotions and motivation, and is also a key to triggering the most authentic thinking processes and behavioral performance.” But what about students who simply dislike games? 

9) there is an extra dot on every main section heading.

Author Response

Reviewer #2:

I read your article with interest. It is well organised, well written and insightful.

  1. The use of the word ‘literacy’ in the title and beginning of the text seems to me to be a bit misleading. Later in your sample you include studies that use games to assess KSAs that are only marginally related to what is meant by literacy. In other words, you seem to be describing your work as narrower in scope than it actually is.

Response: Thank you. Your comments are valuable, and the term ‘literacy’ does tend to raise confusion for readers. We actually focused on all DGBA studies related to student assessment, so we changed the term student literacy assessment (SLA) to student assessment (SA) in the manuscript. Also, we believe that the formulation of knowledge, skills, and abilities (KSAs) does not cover the assessment content of DGBA concerns, so we changed it to knowledge, skills, and affections (KSAs).

  1. I found the opening sentence of the introduction to be rather empty.

Response: Thank you for this comment. We have rewritten this part, hoping that the content is more substantial and easy to understand.

The rewritten content was as follows:

As a core component of psychometric and educational assessment, student assessment (SA) has long been an important research domain of interest to educational researchers. The aim of SA is to comprehend the degree to which students have acquired the knowledge, skills, and affections (KSAs) that are indispensable for full participation in contemporary societies.

  1. I believe 1.1. should be expanded. You make your argument vulnerable to the following objection: “sure, prevalent methods can’t capture a lot of interesting stuff, but they are simple, and the end result would not be very much different.” I think you need to preempt this objection for instance by noticing that students who are at a disadvantage in traditional tests are usually from less privileged backgrounds, which might amplify inequalities, or simply by bringing empirical literature to show that the results of using traditional methods are not highly correlated with the results of using the game based ones that you explore.

Response: Thank you. Based on your comments, we have extended section 1.1 to add the equity issues of the traditional methods. And we mention equity once again in section 1.2 when we elaborate on the advantages of DGBA. Moreover, you are quite right that game assessment results are not highly correlated with traditional methods is exactly one of the conclusions of this study, so we elaborate on it in the discussion section.

The added content was as follows:

Section1.1: The third problem is that students who are at a disadvantage on traditional tests might be from less privileged backgrounds, which potentially amplify inequalities.

Section1.2: Third, designing a game that is accessible and understandable to most students may somewhat enhance the equality among students.

Section4.1.4: The results of those studies (without supervised modeling) showed that more than 60% of the correlation coefficients were below 0.6. One possible reason is that the results of traditional tests may be influenced by students’ backgrounds. Digital games, however, are deeply integrated into most students’ lives, and well-designed games might somewhat enhance the fairness of assessment, thus leading to differences in the results of the two methods.

  1. More should be said about the cons of using game based methods.

Response: Thank you. We have added some disadvantages of the DGBA method in the recommendations for future research in section 4.2 to raise alarms for future research.

The added content was as follows:

Although DGBA has been widely used, future studies should be alert to the potential disadvantages of applying digital games for SA. For example, students of different genders may have innate differences in gaming skills [53], and how to eliminate such gaps by improving game design or through instructional interventions is important for further consideration. Another issue is the gaming experience. Some students play games frequently so that they can easily grasp the mechanisms and techniques of a new game and thus may outperform those with less experience [54]. Future studies should emphasize these individual differences related to game playing. Otherwise, DGBA, despite weakening students’ backgrounds in SA, risks introducing new inequities.

The new citations are as follows:

[53] Kinzie, M.B.; Joseph, D.R. Gender differences in game activity preferences of middle school children: implications for educational game design. Educational Technology Research and Development. 2008, 56, 643-663.

[54] Breuer, J.; Bente, G. Why so serious? On the relation of serious games and learning. Journal for Computer Game Culture. 2010, 4, 7-24..

  1. 4, l. 154 reads “First, the database and was selected”, so there is something missing.

Response: Thank you. We have rewritten as follows:

First, the database was selected, and a set of search terms was used for fixed queries.

  1. You should clarify what a “non English publication” is.

Response: Thank you. ‘Non English publication’ was not used in the screening of the literature and ‘written not in English’ is a more appropriate discription. Therefore we only kept the‘written not in English’.

  1. 7, l. 259 reads “In reference to Acquah et al., this study finally classifies the games into six genres”. I found this sentence cryptic.

Response: Thank you. We have rewritten as follows:

Acquah et al. classified the digital games used for learning into six genres, and the present study refers to this classification. The six genres are adventure, simulation, strategy, role-playing, educational, and puzzle.

  1. 15, ll. 513-518 reads “This is because entertainment is virtually guaranteed to stimulate students’ emotions and motivation, and is also a key to triggering the most authentic thinking processes and behavioral performance.” But what about students who simply dislike games?

Response: Thank you. We did not take into account the students who do not like games. We have rewritten as follows:

Another suggestion is that future research should break the educational game bottleneck, take into account other game genres, and put more work into creating game situations that strike a balance between enjoyment and education. This is because for most students, entertainment is guaranteed to stimulate students’ emotions and motivation, and is also a key to triggering the most authentic thinking processes and behavioral performance.

  1. There is an extra dot on every main section heading.

Response: Thank you. We have modified.

Reviewer 3 Report

General Comments:

In the document, the authors present a systematic review of the assessment based on digital games for students' literacy. Digital games are excellent alternatives for the development of literacy since we live in a society in the constant evolution of information and communication technologies where students belong to a digital generation. In this context, digital game-based learning is also about fun, engagement, and joining meaningful learning with interactive entertainment in a powerful medium that attracts students' attention. However, it is necessary to discuss the use of digital games in the literacy process as a pedagogical tool. It is necessary to discuss and analyze the possibilities of digital games as a pedagogical resource. For example, one can ask, among other questions: to what extent do digital games in literacy stimulate the learning of children in the initial process of reading and writing?

I make this brief introduction to highlight the pertinence of the work presented. I think it is a relevant, current, and interesting topic for researchers, students, and professionals working or interested in the area. In the study, the authors started from 1,553 researched articles and selected 50 for the final analysis.

The document is organized as follows. 1. Introduction; 2.Methodology; 3. Results; 4. Discussion; and, 5. Conclusion. I understand that the sections are sufficient to organize a scientific article. I think the document has a good structure and good wording. The reading was enjoyable for me, the results presented are very rich and consistent. I make a reservation in relation to the section that deals with the conclusion, I think that its wording should be improved.

Below I leave some specific comments related to the parts that make up the document, individually.

Specific comments:

Title:

The title is of adequate length, with 10 words, (A Systematic Review of Digital Game-Based Assessment of Students’ Literacy).

It should be considered that the title is the first component to be read in an article and, therefore, it is the most important sentence in it. We must not forget that the reader can select our article for reading based on the assigned title and, therefore, it must reflect its content, be concise, and include the most relevant terms of the objective of the work.

Regarding writing, the title should answer three fundamental questions: 1. What was done? 2. What was it made about? 3. Where was it made? I don't think it answers the questions clearly.

It should be borne in mind that the title is the first component of an article that will be read and, therefore, it is the most important sentence in it. It should not be forgotten that the reader can select our article for reading based on the assigned title and, therefore, this should reflect its content, be concise, and include the most relevant terms of the objective of the work.

Regarding writing, the title should answer three fundamental questions: 1. What was done? 2. What was it made about? 3. Where was it made? I think it answers the questions.

Abstract

The “abstract” is very important because of its significant use in electronic databases. The text presented in the first version of the document had 217 words, distributed in 10 sentences and presenting an average of 21.7 words/sentence. I think the number of words is adequate, but the average number of words used in the sentences is a little high and could make it difficult to read, for someone who is not very familiar with the topic. For example, for a better reader experience, it is recommended to write more objective sentences. According to the Oxford Guide for Writing (2020):

• Phrases up to 12 words are easy

• Phrases of 13-17 words are acceptable

• Sentences with 18-25 words are difficult

• Sentences longer than 25 words are very difficult

Regarding construction, I think that the abstract still has structural problems and, in my opinion, does not satisfactorily present the necessary items for this part of the document. Namely: “What was studied” (Introduction); “How the study was carried out” (Materials and Methods); “What was found” (Results) and “What does it mean” (Conclusion). I understand that the abstract could be improved if it were written based on the suggestions mentioned above. Another observation that I consider pertinent is that the summary must be written in the past tense, except for the last paragraph or the concluding sentence. Which does not occur in the part of the text. Therefore, I am of the opinion that the abstract needs to be rewritten.

Keywords:

The authors present 4 (digital game-based assessment; student literacy assessment; review; assessment methods). I think it's a suitable number. However, it is desirable that the order of presentation of the keywords be from the most comprehensive to the most specific, which does not occur in the document.

Keywords are a tool to help crawlers and search engines find relevant articles. If database search engines can find your journal manuscript, readers can find it too. This will increase the number of people reading your manuscript and will likely lead to more citations.

Introduction:

This section aims to explain why the study is needed and why it should be published. It should summarize the justification of the study and should arouse the interest of the editor and the reader. The introduction should clearly establish the nature and scope of the problem studied. In other words, the Introduction aims to:

·         Explain the general problem;

·         Define the research problem;

·         Present the background on which the study is based;

·         Define the objectives of the study.

On the other hand, a systematic review article aims to examine published literature on a given topic and put it into perspective to answer a well-defined and structured question. The fundamental objective of a systematic review article is to try to identify what is known about the subject, what has been investigated, what are the most prominent advances in a given period of time and what aspects remain unknown, which allow answering the question of search. I think the section presented satisfactory wording and structuring for this part of the document. However, it should be noted that the definition of objectives and research questions to be investigated are very important topics in this part of the document. Regarding the questions researched, the authors indicate four, which in my view are adequate, however, they do not make the main objective of the review explicit. Although they worked on the problematization.

Methodology

This should describe the method of location, selection, and evaluation of primary studies. It should also be explained how, with what criteria, and which papers were selected and reviewed. The sources consulted, the strategies for carrying out the research, and the period of time for carrying out the study must be described. Clearly indicate the number of references included and excluded by phase of the work, describe the type of studies that were considered for analysis, and the method of classification, codification, and evaluation of the quality of the information.

To guide the systematic review, the authors used the PRISMA (Preferred Reporting Items for Systematic Reviews and Meta-Analyses) recommendation, which aims to help authors improve the reporting of systematic reviews and meta-analyses, which, although it was originally conceived to target -analyses of randomized clinical trials, can also meet several conceptual and practical advances in the science of systematic reviews.

The main steps for carrying out a systematic review go through the definition of search terms and identification of databases and search engines, as well as journals that can be accessed manually and query the selected search terms.

The authors defined the following search terms:

• Search sets related to digital games: “computer games” OR “video games” OR “serious games” OR “digital learning games” OR “digital education games” OR “digital games” OR “online games” OR “Internet games” OR “game-based” OR “game-assisted” OR “game-enhanced” OR “gameplay” OR “epistemic games”.

• Search sets related to assessment: “assessment” OR “evaluation” OR “evaluate” OR “evaluating” OR “measure” OR “measuring” OR “measurement”.

The two search sets were connected by the AND operator.

I think they are adequate, as the number of documents initially returned was 1,553 studies. Regarding the database, the authors used only the Web of Science (WOS). I understand that WOS is an official database that covers a wide range of scientific citations, however, I am of the opinion that if other databases were used, the work would also be much richer, for example, Scopus, ProQuest, ScienceDirect, IEEE Xplore digital library, ACM Digital Library, Springer, among others.

Regarding the filters for inclusion and exclusion (Table 1: The inclusion and exclusion criteria of this study), I think they are adequate. Therefore, the resulting articles (50) were representative and the classification of these indicated that the bias of the publications was related to the objective of the work.

Results

The epilogue of a survey is to show the results. I think the results are well-presented and consistent. In a systematic review, it is expected that the results presented can describe relevant information from emblematic studies and make an objective critique of these studies. I think the authors have worked on this part of the document well and have also used tables and illustrations to analyze and present the results, making them clearer and simpler to interpret.

Discussion

Regarding the Discussion, the document should describe the current knowledge of the research problem and identify possible gaps in the knowledge base. It would be important to explain the differences in the primary studies of the reviewed articles (design, biases, results, etc.) and the discussed and argued synthesis of the results. I think it could have been better worked on in this aspect.

Conclusion

People usually read the abstract, introduction, and conclusion. Although systematic reviews are very useful and important tools for summarizing information, it is not always possible to summarize the results of primary studies, so the conclusion must be meticulous and cautious, only concluding based on the results analyzed and obtained from the work of research and not include information obtained from studies other than your review.

I think that the section presented by the authors is very brief (140 words) and fell short. I am of the opinion that it could be improved.

References

The authors provide a list of 50 references that are all cited in the document. I think the number of references is adequate for the document's profile. Regarding the relevance of the references, 52% of the references are publications made five years ago or less. Of the remaining 48%: 28% are between five and ten years old and 20 are publications over ten years old. I think it is an average indicator, in relation to the profile of the document.

Graphic Elements

Figures, tables and charts are intended to communicate information visually and quickly. The authors present 8 figures and 6 tables, which I think is a good number for the document. They are duly identified and help to read and understand the document. 

Author Response

Reviewer #3:

  1. Regarding writing, the title should answer three fundamental questions: 1. What was done? 2. What was it made about? 3. Where was it made? I think it answers the questions.

Response: Thank you. In response to your comments, we have reformulated the title to highlight issues such as what this study did, what was it made about, etc. The new title is as follows:

Beyond the Traditional: A Systematic Review of Digital Game-Based Assessment for students’ knowledge, skills, and affections

  1. Regarding construction, I think that the abstract still has structural problems and, in my opinion, does not satisfactorily present the necessary items for this part of the document. Namely: “What was studied” (Introduction); “How the study was carried out” (Materials and Methods); “What was found” (Results) and “What does it mean” (Conclusion). I understand that the abstract could be improved if it were written based on the suggestions mentioned above. Another observation that I consider pertinent is that the summary must be written in the past tense, except for the last paragraph or the concluding sentence. Which does not occur in the part of the text. Therefore, I am of the opinion that the abstract needs to be rewritten.

Response: Thank you. Following your comments, we have rewritten the abstract and adjusted the structure so that it meets the framework you presented. Your comments are very useful to us. The rewritten abstract is shown below:

Traditional methods of student assessment (SA) include self-reported surveys, standardized tests, etc. These methods are widely regarded by researchers as inducing test anxiety. They also ignore students’ thinking processes and are not applicable to the assessment of higher-order skills. Digital game-based assessment (DGBA) is thought to address the shortcomings of traditional assessment methods. Given the advantages of DGBA, an increasing number of empirical studies are working to apply digital games for SA. However, there is a lack of any systematic review of DGBA studies. In particular, very little is known about the characteristics of the games, the content of the assessment, the methods of implementation, and the distribution of the results. This study examined the characteristics of DGBA studies, and the adopted games on SA in the past decade from different perspectives. A rigorous systematic review process was adopted in this study. First, the Web of Science (WOS) database was used to search the literature on DGBA published over the last decade. Then, 50 studies on SA were selected for subsequent analysis according to the inclusion and exclusion criteria. The results of this study found that DGBA has attracted the attention of researchers around the world. The participants of the DGBA studies were distributed across different educational levels, but the number of participants was small. Among all game genres, educational games were the most frequently used. Disciplinary knowledge is the most popular SA research content. Formative assessment modeling with process data and summative assessment using final scores were the most popular assessment methods. Correlation analysis was the most popular analysis method to verify the effectiveness of games on SA. However, many DGBA studies have reported unsatisfactory data analysis results. For the above findings, this study further discussed the reasons, as well as the meanings. In conclusion, this review showed the current status and gaps of DGBA in the SA application; directional references for future research of researchers and game designers are also provided.

  1. The authors present 4 (digital game-based assessment; student literacy assessment; review; assessment methods). I think it's a suitable number. However, it is desirable that the order of presentation of the keywords be from the most comprehensive to the most specific, which does not occur in the document.

Response: Thank you. We redefined four keywords to ensure they reflected the core content of the document while not occurring in the document. The new keywords are as follows:

Keywords: assessment methodologies; digital games; 21st century skills; media in education

  1. Regarding the questions researched, the authors indicate four, which in my view are adequate, however, they do not make the main objective of the review explicit. Although they worked on the problematization.

Response: In response to your comments, we have added relevant content with the aim of enhancing the link between the main objectives of the review and the research questions. The new content is shown as follows:

The objective of this review is to fill in the gaps of previous review related to DGBA, and to make suggestions for future DGBA studies. In particular, we focus on the studies in which students were the subject of assessment. We further present a comprehensive overview of the current status and shortcomings of DGBA studies in terms of the distribution of participants, the characteristics of the games, the application of the assessment methods, and the results of the data analysis. Based on the findings of our review, recommendations for future DGBA studies are further proposed. Specifically, the present study would respond to the following research questions (RQ):

RQ1: What was the overview of the participants in the DGBA studies, including the country regions, number, and educational levels of the participants?

RQ2: What were the characteristics of the games used in the DGBA studies, including the game genres, game platforms, and commercial access to the games?

RQ3: What were the main assessment contents, and what assessment methods were used in the DGBA studies?

RQ4: What data analysis techniques were adopted, and what data analysis results were reported in the DGBA studies?

  1. I think they are adequate, as the number of documents initially returned was 1,553 studies. Regarding the database, the authors used only the Web of Science (WOS). I understand that WOS is an official database that covers a wide range of scientific citations, however, I am of the opinion that if other databases were used, the work would also be much richer, for example, Scopus, ProQuest, ScienceDirect, IEEE Xplore digital library, ACM Digital Library, Springer, among others.

Response: Thank you. This study is an extension of our previous brief review. In our previous study we used the WOS database, and to ensure that the findings of the two can be compared and echoed, we still used WOS as the search database. Compared to the previous review, we added new literature in 2022, as shown in Figure 1 in the manuscript. Morever, in WOS, we retrieved more than 1500 relevant literatures and finally screened 50 literatures that met the inclusion criteria. These literatures are sufficient for a systematic review to reveal the current status and gaps of researches in the field of DGBA.

  1. The epilogue of a survey is to show the results. I think the results are well-presented and consistent. In a systematic review, it is expected that the results presented can describe relevant information from emblematic studies and make an objective critique of these studies. I think the authors have worked on this part of the document well and have also used tables and illustrations to analyze and present the results, making them clearer and simpler to interpret.

Response: Thank you.

  1. Regarding the Discussion, the document should describe the current knowledge of the research problem and identify possible gaps in the knowledge base. It would be important to explain the differences in the primary studies of the reviewed articles (design, biases, results, etc.) and the discussed and argued synthesis of the results. I think it could have been better worked on in this aspect.

Response: Thank you. Your comments are very valuable. We really need to go more into the gap analysis in the discussion section. In order to show the systematic discussion of the results we have derived for each research question, we have adjusted the discussion framework by adding four sub-headings with the four research questions as content. Also, we have added some discussion of the results. The new contents are shown as follows:

4.1.1. The overview of the participants in the DGBA studies

For RQ1, this study first found that DGBA research in the field of SA has covered six continents; this clearly shows growing interest from DGBA researchers at a global level. Of all continents, North America has conducted the most studies, followed by Europe and Asia. This study went further and counted the specific countries, and the studies conducted in the United States were multiple times higher than in other countries, accounting for more than 40% of the total number of studies. Although DGBA studies are widely distributed, the United States accounts for the majority, which proves that current DGBA research is inadequate and should attract the attention of researchers in more countries in the future.

4.1.3. The assessment contents and methods used in the DGBA studies

For RQ3, echoing the largest proportion of educational games among the game genres discussed above, this study found that discipline-specific knowledge was the most popular assessment content for researchers. This may be attributed to the fact that assessment of knowledge of a particular discipline is the type most in demand in school education. In addition, educational games used to assess discipline-specific knowledge are less difficult to produce. Notably, an essential advantage of DGBA is the creation of authentic situations that stimulate complex problem-solving performance, with the game results used to measure students’ higher-order skills. However, the shortcoming of previous DGBA studies is that a large portion of the studies focused on relatively low-order, discipline-specific knowledge. Therefore, the assessment of higher-order contemporary competencies needs to be reinforced to fully exploit the advantages of DGBA over traditional assessment methods.

4.1.4. The data analysis techniques and results reported in the DGBA studies

The results of the data analysis were not highly satisfactory. In the supervised regression models, the R2 indicator only reached a maximum of 0.431; the accuracy of the supervised classification models only reached a maximum of about 0.7, and half of the correlation coefficients between the prediction results of the supervised models and the external tests were below 0.5. The results of those studies (without supervised modeling) showed that more than 60% of the correlation coefficients were below 0.6. One possible reason is that the results of traditional tests may be influenced by students’ backgrounds. Digital games, however, are deeply integrated into most students’ lives, and well-designed games might somewhat enhance the fairness of assessment, thus leading to differences in the results of the two methods. On the other hands, some DGBA studies have shown that transforming game data into constructs is a difficult procedure that typically yields low-performing results [51,52]. Regardless of the merits or demerits of the game design, most previous DGBA studies have been exploratory, and they aim to demonstrate the potential of DGBA to some extent. Therefore, researchers have considered that, although some studies had reported relatively low-performing results, most of these results were statistically significant. Also, these results are beneficial for future advances in the research field of DGBA [51].

  1. People usually read the abstract, introduction, and conclusion. Although systematic reviews are very useful and important tools for summarizing information, it is not always possible to summarize the results of primary studies, so the conclusion must be meticulous and cautious, only concluding based on the results analyzed and obtained from the work of research and not include information obtained from studies other than your review.I think that the section presented by the authors is very brief (140 words) and fell short. I am of the opinion that it could be improved.

Response: Thank you. In response to your comments, we have added meticulous and cautious findings to the conclusion section, which makes the conclusion section more substantial. The new conclusion is shown as follows:

This study aims to investigate DGBA studies, and the games adopted for SA from various perspectives over the past decade. This comprehensive review presents surprising findings on the use of DGBA in the field of SA. The analysis reveals that DGBA has gained global attention from researchers, especially in the United States. The reviewed studies encompassed participants from various educational levels, with K-12 receiving the most attention. Nonetheless, the DGBA studies exhibited a tendency towards small sample sizes. Educational games, which are less entertaining, emerged as the most frequently used game genre. Disciplinary knowledge was the primary focus of assessment in the reviewed studies, as opposed to 21st century higher-order skills. The most common assessment methods included formative assessment modeling with process data and summative assessment using final scores. To validate the effectiveness of DGBA on SA, diverse data analysis methods were utilized, with correlational analysis being the most widespread. However, the results of data analysis from a considerable number of studies reported unsatisfactory efficacy of DGBA for SA. In conclusion, this study sheds light on the current status and gaps of DGBA studies in SA and offers directions for future research.

  1. The authors provide a list of 50 references that are all cited in the document. I think the number of references is adequate for the document's profile. Regarding the relevance of the references, 52% of the references are publications made five years ago or less. Of the remaining 48%: 28% are between five and ten years old and 20 are publications over ten years old. I think it is an average indicator, in relation to the profile of the document.

Response: Thank you.

  1. Graphic Elements. Figures, tables and charts are intended to communicate information visually and quickly. The authors present 8 figures and 6 tables, which I think is a good number for the document. They are duly identified and help to read and understand the document.

Response: Thank you.

Reviewer 4 Report

The study that is proposed and analysed is, in my view, of great value for science. However, I believe that there are aspects that could be improved:

1.- The objective of the work, explicitly named, appears on line 549, in the conclusions. In my opinion, it should appear in the abstract.  

It would be very interesting if the research questions RQ1,2,3,4 were answered in the conclusions and/or in the discussion. 

Author Response

Reviewer #4:

The study that is proposed and analysed is, in my view, of great value for science. However, I believe that there are aspects that could be improved:

  • The objective of the work, explicitly named, appears on line 549, in the conclusions. In my opinion, it should appear in the abstract.

Response: Thank you. Based on your comments, we put this sentence in the abstract. The new abstract is as follows:

Traditional methods of student assessment (SA) include self-reported surveys, standardized tests, etc. These methods are widely regarded by researchers as inducing test anxiety. They also ignore students’ thinking processes and are not applicable to the assessment of higher-order skills. Digital game-based assessment (DGBA) is thought to address the shortcomings of traditional assessment methods. Given the advantages of DGBA, an increasing number of empirical studies are working to apply digital games for SA. However, there is a lack of any systematic review of DGBA studies. In particular, very little is known about the characteristics of the games, the content of the as-sessment, the methods of implementation, and the distribution of the results. This study examined the characteristics of DGBA studies, and the adopted games on SA in the past decade from different perspectives. A rigorous systematic review process was adopted in this study. First, the Web of Science (WOS) database was used to search the literature on DGBA published over the last decade. Then, 50 studies on SA were selected for subsequent analysis according to the inclusion and ex-clusion criteria. The results of this study found that DGBA has attracted the attention of researchers around the world. The participants of the DGBA studies were distributed across different educa-tional levels, but the number of participants was small. Among all game genres, educational games were the most frequently used. Disciplinary knowledge is the most popular SA research content. Formative assessment modeling with process data and summative assessment using final scores were the most popular assessment methods. Correlation analysis was the most popular analysis method to verify the effectiveness of games on SA. However, many DGBA studies have reported unsatisfactory data analysis results. For the above findings, this study further discussed the reasons, as well as the meanings. In conclusion, this review showed the current status and gaps of DGBA in the SA application; directional references for future research of researchers and game designers are also provided.

  • It would be very interesting if the research questions RQ1,2,3,4 were answered in the conclusions and/or in the discussion.

Response: Thank you. Your comments are very valuable. In order to show the systematic discussion of the results we have derived for each research question, we have adjusted the discussion framework by adding four sub-headings with the four research questions as content. The adjusted structure of the discussion section is shown as follows:

4.1.1. The overview of the participants in the DGBA studies

For RQ1, this study first found that DGBA research in the field of SA has covered six continents; this clearly shows growing interest from DGBA researchers at a global level. Of all continents, North America has conducted the most studies, followed by Europe and Asia. This study went further and counted the specific countries, and the studies conducted in the United States were multiple times higher than in other countries, accounting for more than 40% of the total number of studies. Although DGBA studies are widely distributed, the United States accounts for the majority, which proves that current DGBA research is inadequate and should attract the attention of researchers in more countries in the future.

……

4.1.2. The characteristics of the games used in the DGBA studies

For RQ2, according to the findings of this study, educational games have emerged as the most commonly used game genre in DGBA studies, which is consistent with Acquah’s research [30]. This outcome is further corroborated by the observation that the assessment content has largely centered on discipline-specific knowledge. The structure of educational games is generally more well-defined, with a greater emphasis on educational content and a lower emphasis on entertainment value. Furthermore, when compared to other game genres, educational games are comparatively easier to develop. These characteristics make educational games a suitable choice for assessing students’ discipline-related knowledge. However, researchers have pointed out that a narrow focus on knowledge related to a specific discipline through simple game design may create a difficulty in making the game appealing to students [23]. Among all game genres, the number of studies that adopted simulation games was the lowest. This may be due to the fact that simulation games require a high degree of reproduction of real-life scenarios, usually combined with virtual reality techniques. Such games are extremely difficult to develop.

……

4.1.3. The assessment contents and methods used in the DGBA studies

For RQ3, echoing the largest proportion of educational games among the game genres discussed above, this study found that discipline-specific knowledge was the most popular assessment content for researchers. This may be attributed to the fact that assessment of knowledge of a particular discipline is the type most in demand in school education. In addition, educational games used to assess discipline-specific knowledge are less difficult to produce. Notably, an essential advantage of DGBA is the creation of authentic situations that stimulate complex problem-solving performance, with the game results used to measure students’ higher-order skills. However, the shortcoming of previous DGBA studies is that a large portion of the studies focused on relatively low-order, discipline-specific knowledge. Therefore, the assessment of higher-order contemporary competencies needs to be reinforced to fully exploit the advantages of DGBA over traditional assessment methods.

……

4.1.4. The data analysis techniques and results reported in the DGBA studies

For RQ4, in term of those studies that used formative assessment modeling with process data, this review finds that the previous studies mostly constructed multiple supervised models, with linear regression, random forest, and support vector machine being the most popular algorithms. Another striking finding is that 58% of the studies adopted the technique of correlation analysis. The reason for this is that the general purpose of a DGBA study is to verify the validity of the game used to assess the content of interest. Therefore, the most common practice is to analyze the correlation between the final game scores, the process data, the indicators calculated from process data, or the predicted results of the supervised models and the results of the external tests’ criterion.

……
